# Short-term effects of cold spells on hospitalisations for acute exacerbation of chronic obstructive pulmonary disease: a time-series study in Beijing, China

Yanbo Liu,[1] Yuxiong Chen,[1] Dehui Kong,[1] Xiaole Liu,[1] Jia Fu ![ORCID],[1] Yongqiao Zhang,[1] Yakun Zhao,[1] Zhen'ge Chang,[1] Xiaoyi Zhao,[2] Kaifeng Xu,[1] Chengyu Jiang,[3] Zhongjie Fan ![ORCID] [1]

► Prepublication history and supplemental material for this paper are available online. To view these files, please visit the journal online (http://dx.doi.org/10.1136/bmjopen-2020-039745).

YL and YC contributed equally.

For numbered affiliations see end of article.

**Correspondence to**
Professor Zhongjie Fan;
Fanzhongjie@pumch.cn

## ABSTRACT

**Objectives** Our work aimed at exploring the relationship between cold spells and acute exacerbation of chronic obstructive pulmonary disease (AECOPD) hospitalisations in Beijing, China, and assessing the moderating effects of the intensities and the durations of cold spells, as well as identifying the vulnerable.

**Design** A time-series study.

**Setting** We obtained time-series data of AECOPD hospitalisations, meteorological variables and air quality index in Beijing, China during 2012–2016.

**Participants** All AECOPD hospitalisations among permanent residents in Beijing, China during the cold seasons (November–March) of 2012–2016 were included (n=84 571).

**Primary and secondary outcome measures** A quasi-Poisson regression with a distributed lag model was fitted to investigate the short-term effects of cold spells on AECOPD hospitalisations by comparing the counts of AECOPD admissions during cold spell days with those during non-cold spell days.

**Results** Cold spells under different definitions were associated with increased risk of AECOPD hospitalisations, with the maximum cumulative relative risk (CRR) over 3 weeks (lag0–21). The cumulative effects at lag0–21 increased with the intensities and the durations of cold spells. Under the optimal definition, the most significant single-day relative risk (RR) was found on the days of cold spells (lag0) with an RR of 1.042 (95% CI 1.013 to 1.072), and the CRR at lag0–21 was 1.394 (95% CI 1.193 to 1.630). The elderly (aged ≥65) were more vulnerable to the effects of cold spells on AECOPD hospitalisations.

**Conclusion** Cold spells are associated with increased AECOPD hospitalisations in Beijing, with the cumulative effects increased with intensities and durations. The elderly are at particular risk of AECOPD hospitalisations triggered by cold spells.

## INTRODUCTION

The Intergovernmental Panel on Climate Change has predicted that human activities

### Strengths and limitations of this study

► This study was the first to examine the association between cold spells and acute exacerbation of chronic obstructive pulmonary disease (AECOPD) hospitalisations in China.

► The study assessed the effects of cold spells under different definitions on AECOPD hospitalisations to find out the optimal cold spell definition on the issue.

► The ecological design cannot imply causality definitely, while limited information on individual-level factors and inevitable exposure measurement errors may lead to bias.

► The data from one specific city limited the extrapolation of the findings.

and global climate change will cause variations in frequency, intensity and duration of many extreme weather events, including heatwaves and cold spells.[1] Although the amount of cold spells may decrease over most land areas due to global warming, a few recent studies found that the persistent shift of the Arctic polar vortex and the Arctic amplification associated with global warming could lead to increased extremely cold events in mid-latitudes.[2 3] Over the last few years, the impacts of cold spells on human health have gained growing attention from the public. Many studies have reported positive relationships between cold spells and mortality,[4–6] while the impacts of cold spells on hospital visits or admissions are underexamined.

Chronic obstructive pulmonary disease (COPD) is one of the common respiratory diseases characterised by poorly reversible limitation of airflow.[7] Owing to its high prevalence, morbidity, mortality and economic burden globally, COPD has been an important

public health concern and will remain a huge challenge for healthcare practitioners in the foreseeable future.[8] Thus, it is crucial to identify the risk factors of COPD to improve strategies on prevention and intervention. Given the projected climate change, extreme temperature events potentially pose threats to patients with COPD. Many epidemiological studies have indicated that COPD has higher rates of exacerbation and hospitalisation with lower temperatures.[9–13] We hypothesised that cold spells, defined as prolonged periods of extremely cold weather, may be more detrimental to patients with COPD and could cause more hospitalisations for acute exacerbations of chronic obstructive pulmonary disease (AECOPD).[14] However, few studies have been carried out on the association between cold spells and AECOPD hospitalisations.[15]

As the world's largest country by population, China shoulders the enormous burden of COPD. A national cross-sectional study from 2012 to 2015 showed that the prevalence of COPD among Chinese adults aged 20 years and older was 8.6% (an estimated 99.9 million patients with COPD).[16] On the other hand, with most areas located in mid-latitudes, China has experienced several severe cold spells in recent years. The cold spells in 2008 resulted in a significantly higher all-cause mortality in subtropical China and estimated losses exceeding $22.3 billion.[17 18] Moreover, the public now has a better perception of the potential risks of extreme temperatures in China, especially those with chronic conditions.[19] Since no relevant studies have been reported in China, it is of great value to assess the association between cold spells and AECOPD hospitalisations to build prevention and adaption strategies suitable to local conditions (eg, climate type, sociodemographic status of residents), which may be different from other regions.

Cold spells have been defined differently due to the heterogeneity of climate and people's adaptive capacities in different regions. Previous studies suggested that the effects of cold spells varied by different cold spell characteristics and individual-specific factors.[20 21] We have three main objectives in this work: (1) to illuminate the short-term effects of cold spells on the risk of hospitalisations for AECOPD with time-series methods; (2) to investigate the effect modification of cold spell intensities and durations by fitting different definitions and to explore the optimal cold spell definition in this region; and (3) to identify potentially vulnerable populations through stratified analyses. The results could help better understand the relationship between extremely cold events and AECOPD hospitalisations, and provide scientific evidence in policymaking for local prevention and intervention of AECOPD.

## MATERIALS AND METHODS
### Data collection
Beijing, the capital of China, is located in the northern part of China (39°56'N, 116°20'E). The area covers 16 410.54 km², with more than 21 million population in 2016. Beijing has a typical semihumid continental monsoon climate with four distinctive seasons.

Daily hospitalisations for AECOPD from 1 January 2012 to 31 December 2016 were collected from the Beijing Public Health Information Center (http://www.phic.org.cn/). All government and private hospitals at the secondary or tertiary level in Beijing are required to submit their discharge records to the database.[22 23] Each record consists of the following information: admission date, discharge date, age, gender, address, diagnosis and International Classification of Diseases 10th revision (ICD-10) diagnostic code. We excluded records with missing or wrong information on residential addresses. Only those among Beijing residents admitted to hospitals with AECOPD as the primary discharge diagnosis (ICD-10: J44) were included in the study.[23] All data for the analysis were anonymous at collection.

We collected the daily 2012–2016 meteorological data in Beijing from the China Meteorological Data Sharing Service System, including daily mean temperature (°C), daily mean relative humidity (%) and daily mean air pressure (hPa). For the same period, the daily air quality index (AQI) was obtained from the China National Environmental Monitoring Centre. The AQI value denotes the maximum value of individual air quality indexes of six monitored air pollutants (particulate matter with aerodynamic diameter <2.5 µm, particulate matter with aerodynamic diameter <10 µm, nitrogen dioxide, sulfur dioxide, carbon monoxide and ozone). Considering the impact of influenza viral infections on AECOPD,[13 24] we also accessed data on virological surveillance from the Chinese National Influenza Center (CNIC) (http://www.chinaivdc.cn/cnic/). The CNIC monitors the activity of seasonal influenza viruses in China and reports weekly positive rates of influenza isolations in the northern and southern parts separately. In this study, the onset of influenza epidemics (a binary variable representing days with relatively high influenza episodes)[25 26] was defined as when the proportion of isolates positive for influenza in any given week exceeded 30% of the maximum weekly positive isolation rate in the whole surveillance season (influenza surveillance season was defined from the 27th week of the previous year to the 26th week of the following year) in northern China.[27]

### Cold spell definitions
The definition of cold spells varied across the research field due to distinct climatic features and temperature variations in different regions. As to the prior studies, cold spells were usually defined based on temperature thresholds and durations.[4 15 28] Instead of specific temperatures as thresholds, percentiles of temperature were shown to have a better model fit according to quasi-Poisson Akaike information criterion (Q-AIC).[6] Moreover, some researchers suggested that daily mean temperature is superior to the minimum or maximum temperature as an indicator to define cold spells because it reflects the exposure throughout the day rather than a short

period.[20 29] Therefore, we defined cold spell episodes as days when the daily mean temperature was at or below the 10th, 5th or 3rd percentile for at least 2, 3 or 4 consecutive days of the study period.[21] To avoid possible biases caused by a few extreme summer events, we restricted the study period to the five coldest adjacent months (from November of the previous year to March of the following year) for each year.[6 15 20] Cold spell was treated as a dichotomous variable, with a value of 1 during the cold spell period. Statistical analyses were performed separately for each definition of cold spells.

## Statistical methods

In the analyses, the dependent variable was the number of daily AECOPD hospitalisations following a quasi-Poisson distribution. Hence, we adopted a distributed lag model (DLM)[30] with a quasi-Poisson generalised linear regression model. To investigate the effects of cold spells on AECOPD hospitalisations, we compared the counts of AECOPD admissions during cold spell days with those during non-cold spell days, after adjusting for relative humidity, atmospheric pressure, AQI, seasonality, long-term trends, statutory holiday, influenza epidemics and day of the week. The Q-AIC was employed to choose the optimal cold spell definition and df. The model was established as follows:

$$Y_t \sim \text{quasiPoisson}\left(\mu_t\right)$$

$$\text{Log}\left(\mu_t\right) = \alpha + \text{cb}\left(CS_t, \text{ lag}, \text{df} = 3\right) +$$
$$\text{ns}\left(RH_t, \text{ df} = 3\right) + \text{ns}\left(AP_t, \text{ df} = 3\right) +$$
$$\text{ns}\left(AQI_t, \text{ df} = 3\right) + \text{ns}\left(Time_t, \text{df} = 3/\text{per year}\right) + \gamma DOW_t +$$
$$\delta Holiday_t + \nu Influenza_t,$$

where t is the day of observation; $Y_t$ is the expected number of hospitalisations for AECOPD on day t; $\alpha$ is the intercept; $CS_t$ denotes the cold spells on day t (0=non-cold spell days and 1=cold spell days); and cb represents the cross-basis function, including a linear function and a natural cubic spline function with 3 df to assess the linear and lagged effects of the cold spells separately. We fitted a lag structure up to 21 days in the models to completely capture the flexible lagged effects of cold spells exposure. ns refers to the natural cubic spline function. ns with 3 df was applied for the mean relative humidity ($RH_t$), mean atmospheric pressure ($AP_t$) and air quality index ($AQI_t$), respectively. ns with 3 df per year was used to control the seasonality and long-term trends. $DOW_t$ is a categorical variable indicating the day of the week on day t, and $\gamma$ is the coefficient. $Holiday_t$ is a binary variable (0=non-statutory holiday and 1=statutory holiday) and $\delta$ is the coefficient. $Influenza_t$ is a dichotomous variable with the value of '1' for the influenza epidemic on day t, and $\nu$ is the coefficient. The statistical methods, maximum lag days and confounding factors included in the model were commonly used and described in previous publications.[4 17 20 21 31 32]

To observe the variation trend of lagged effects, we calculated the single-day lagged effects (from lag0 to lag21) and cumulative effects (lag0, lag0–7, lag0–14 and lag0–21) of cold spells on AECOPD hospitalisations, respectively. To identify the susceptible subpopulations for more targeted public health interventions, we further conducted subgroup analyses to investigate the potential modification effects by gender (male and female) and age (0–64 years old and ≥65 years old) under the optimal definition of cold spells. The statistical differences of the risk estimates between the subgroups were examined by Z-test with the following equation:

$$Z = \left(E_1 - E_2\right)/\sqrt{\left(SE_1^2 + SE_2^2\right)}$$

where Z represents the Z-test value; $E_1$ and $E_2$ denote the effect estimates of two categories; and $SE_1$ and $SE_2$ are the corresponding standard errors of $E_1$ and $E_2$.

## Sensitivity analysis

We performed the sensitivity analyses by altering the df with 3–5 df per year of the long-term trend, 3–5 df of the relative humidity, 3–5 df of the air pressure, 3–5 df of the AQI and 3–5 df of the lag dimension in the DLM under the optimal definition of cold spells. We used R V.3.6.1 software with the 'dlnm' and 'splines' packages to run the analyses. All statistical tests were two-sided and values of p<0.05 were considered statistically significant.

## Patient and public involvement

Patients were not involved in the development of the research question and outcome measures, study design, or conduct of this study.

## RESULTS
### Data description

Table 1 shows the descriptive statistics of the study population, the meteorological variables and the AQI during the cold seasons (November–March) from 2012 to 2016 in Beijing. There were a total of 84 571 AECOPD hospitalisations throughout the study period. Among these cases, 63.6% were male and 36.4% were female. Of all patients, 83.9% were aged 65 years and above. The average daily mean temperature, relative humidity, air pressure and AQI were 0.9°C (range, −16.0°C to 18.0°C), 46.6% (range, 8.0%–98.0%), 1025.3 hPa (range, 1005.0–1044.0 hPa) and 126.3 (range, 17.0–485.0), respectively.

Table 2 shows the overview information of cold spells under different definitions. More days were defined as cold spell days with higher temperature thresholds and shorter duration. We observed the most cold spell episodes and days in 4 years using the definition of periods of at least 2 days with daily mean temperature below or at the 10th percentile (−6°C). In contrast, there were only two cold spell episodes (10 cold spell days) if we defined cold spells as periods of 4 or more consecutive days when the daily mean temperature was below or at the third percentile (−8°C).

**Table 1** Statistics summary of daily hospitalisations for AECOPD (counts per day), meteorological variables and air quality index during cold seasons (November–March) in Beijing, China, 2012–2016

|  | Total | Mean (SD) | Minimum | P25 | Median | P75 | Maximum |
|---|---|---|---|---|---|---|---|
| AECOPD hospitalisations | 84 571 | 111.7 (41.0) | 22.0 | 78.0 | 113.0 | 141.0 | 226.0 |
| Gender |  |  |  |  |  |  |  |
| Male | 53 829 | 71.1 (25.8) | 12.0 | 49.0 | 75.0 | 89.0 | 158.0 |
| Female | 30 742 | 40.6 (17.7) | 7.0 | 27.0 | 39.0 | 53.0 | 110.0 |
| Age (years) |  |  |  |  |  |  |  |
| 0–64 | 13 600 | 18.0 (8.3) | 2.0 | 12.0 | 18.0 | 23.0 | 43.0 |
| ≥65 | 70 971 | 93.8 (34.3) | 19.0 | 67.0 | 95.0 | 118.0 | 189.0 |
| Environmental variables |  |  |  |  |  |  |  |
| Mean temperature (°C) | – | 0.9 (5.4) | −16.0 | −3.0 | 0.0 | 4.0 | 18.0 |
| Relative humidity (%) | – | 46.6 (20.4) | 8.0 | 30.0 | 43.0 | 61.0 | 98.0 |
| Air pressure (hPa) | – | 1025.3 (6.5) | 1005.0 | 1021.0 | 1026.0 | 1030.0 | 1044.0 |
| Air quality index | – | 126.3 (89.2) | 17.0 | 61.0 | 97.0 | 170.0 | 485.0 |

AECOPD, acute exacerbation of chronic obstructive pulmonary disease; P25, 25th percentile; P75, 75th percentile.;

## Effects of cold spells under different definitions

Figure 1 depicts the lag structures of associations between cold spells under nine different definitions and AECOPD hospitalisations of the total population. All cold spells had an impact on the risk of hospitalisations for AECOPD, and most trends of their lagged effects were non-linear with two patterns. One was that the relative risk (RR) of hospitalisations for AECOPD reached a maximum on the days (lag0) of cold spells, then decreased and remained significant for 10–16 days (lag10–lag16). The other one was that the RR became significant on the 3rd or 4th day (lag3 or lag4) after exposure to cold spells, then gradually reached the maximum at about the 8th day (lag8) and lasted until the 15th–17th days (lag15–lag17).

Table 3 shows the cumulative effects of cold spells on AECOPD hospitalisations under different definitions. For each definition, the cumulative relative risk (CRR) increased with longer cumulative lags, with the highest CRR at lag0–21. Among the nine different definitions, the CRR of cold spells at lag0–21 increased using the definition with longer duration or lower temperature threshold. The maximum CRR over lag0–21 when the temperature threshold was set at the 10th percentile appeared at a duration of ≥4 consecutive days. In comparison, the CRR values at durations of ≥3 consecutive days and ≥2 consecutive days were lower. Likewise, when the duration was set for at least 4 consecutive days, the maximum CRR over lag0–21 appeared at the temperature range defined as ≤3rd percentile, while the CRRs of the temperature threshold at the 5th and 10th percentiles were lower. The lowest Q-AIC value (7769.8) indicating the best model fit was observed in model 3 (table 2). Hence, the optimal cold spell definition was daily mean temperature ≤10th percentile (−6°C) for at least 4 consecutive days during the study period. The optimal model was able to find the most significant single-day lagged effect earliest (lag0) and remained significant until the 14th day (lag14) (figure 1).

**Table 2** Overview information of cold spells under different definitions

| Model | Temperature threshold | Duration (days) | Cold spell episodes | Cold spell days | Non-cold spell days | Q-AIC value |
|---|---|---|---|---|---|---|
| 1 | ≤10% (−6°C) | ≥2 | 17 | 77 | 680 | 7819.6 |
| 2 | ≤10% (−6°C) | ≥3 | 11 | 65 | 692 | 7799.7 |
| 3 | ≤10% (−6°C) | ≥4 | 6 | 50 | 707 | 7769.8 |
| 4 | ≤5% (−7°C) | ≥2 | 8 | 37 | 720 | 7794.3 |
| 5 | ≤5% (−7°C) | ≥3 | 5 | 31 | 726 | 7786.6 |
| 6 | ≤5% (−7°C) | ≥4 | 3 | 25 | 732 | 7782.5 |
| 7 | ≤3% (−8°C) | ≥2 | 7 | 23 | 734 | 7786.5 |
| 8 | ≤3% (−8°C) | ≥3 | 5 | 19 | 738 | 7789.8 |
| 9 | ≤3% (−8°C) | ≥4 | 2 | 10 | 747 | 7804.4 |

Q-AIC, quasi-Poisson Akaike information criterion.

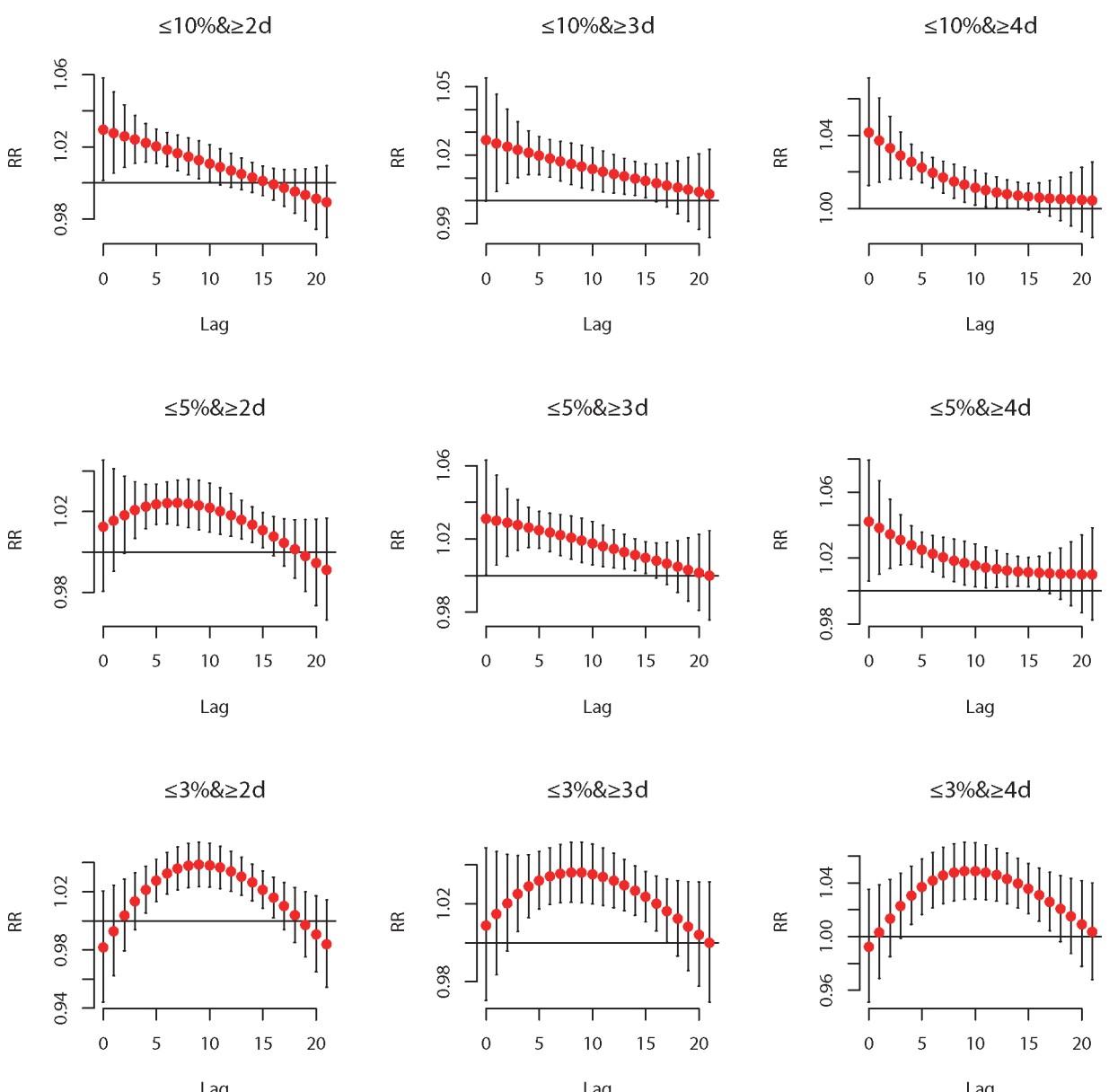

**Figure 1** Lag–response relationships between cold spells under different definitions and AECOPD hospitalisations of the total population in Beijing, 2012–2016. AECOPD, acute exacerbation of chronic obstructive pulmonary disease; RR, relative risk.

Table 4 and figure 2 reveal the results for the subgroup analyses of gender and age based on the optimal cold spell definition. The effects of cold spells were similar between male and female patients, with the most significant single-day lagged effect both occurring at lag0 (Z=0.041, p=0.48). The cumulative effects at lag0–21 of the two genders also differed slightly (Z=−0.730, p=0.23). Additionally, in the subgroups stratified by age, the most significant single-day lagged effect and cumulative effect for people aged ≥65 years were at lag0 and lag0–21, respectively. However, no significant effect of cold spells was observed in those aged 0–64 years. The results of sensitivity analyses indicated that the effect estimates of cold spells under the optimal definition on AECOPD hospitalisations were still robust (see online supplemental tables 1–5).

## DISCUSSION

In this study, we showed that cold spells were associated with increased hospitalisations for AECOPD. The adverse impacts of cold spells varied with durations and intensities. Based on the statistical model fit, the optimal definition of cold spells was daily mean temperature less than or equal to the 10th percentile lasting for at least 4 consecutive days during the study period. The elderly seemed more sensitive to cold spells than the younger, while the susceptibility difference between genders was not noticeable.

Our finding is in accordance with previous studies reporting an excess of AECOPD hospitalisations,[15] emergency visits[33] and mortality[4 6 34] associated with cold spells. For instance, Monteiro et al[15] reported significant effects

**Table 3** Cumulative relative risk of cold spells under different definitions on AECOPD hospitalisations of the total population in Beijing, 2012–2016

| Definition | CRR (95% CI) | | | |
| --- | --- | --- | --- | --- |
| | Lag0 | Lag0–7 | Lag0–14 | Lag0–21 |
| ≤10% and ≥2 days | 1.030 (1.002 to 1.058)* | 1.200 (1.087 to 1.326)* | 1.278 (1.134 to 1.439)* | 1.236 (1.055 to 1.448)* |
| ≤10% and ≥3 days | 1.027 (1.000 to 1.054)* | 1.188 (1.084 to 1.303)* | 1.299 (1.166 to 1.448)* | 1.353 (1.161 to 1.577)* |
| ≤10% and ≥4 days | 1.042 (1.013 to 1.072)* | 1.249 (1.136 to 1.374)* | 1.343 (1.206 to 1.496)* | 1.394 (1.193 to 1.630)* |
| ≤5% and ≥2 days | 1.012 (0.980 to 1.045) | 1.173 (1.055 to 1.304)* | 1.344 (1.188 to 1.519)* | 1.354 (1.141 to 1.608)* |
| ≤5% and ≥3 days | 1.031 (1.000 to 1.063)* | 1.235 (1.114 to 1.369)* | 1.381 (1.220 to 1.563)* | 1.428 (1.206 to 1.692)* |
| ≤5% and ≥4 days | 1.042 (1.006 to 1.079)* | 1.268 (1.132 to 1.421)* | 1.404 (1.234 to 1.598)* | 1.511 (1.262 to 1.809)* |
| ≤3% and ≥2 days | 0.982 (0.944 to 1.021) | 1.113 (0.960 to 1.290) | 1.412 (1.171 to 1.703)* | 1.444 (1.113 to 1.873)* |
| ≤3% and ≥3 days | 1.009 (0.970 to 1.049) | 1.217 (1.051 to 1.411)* | 1.525 (1.264 to 1.841)* | 1.659 (1.278 to 2.154)* |
| ≤3% and ≥4 days | 0.992 (0.951 to 1.035) | 1.201 (0.999 to 1.443) | 1.644 (1.261 to 2.145)* | 1.889 (1.315 to 2.712)* |

*P<0.05.

AECOPD, acute exacerbation of chronic obstructive pulmonary disease; CRR, cumulative relative risk.

of cold spells identified by various indices on COPD hospitalisations with a lagged effect of at least 2 weeks. Several underlying mechanisms may explain for elevated COPD morbidity and mortality attributable to extremely cold events. First, cold exposure has been found to be related to a decline in lung function (forced expiratory volume in 1 s, forced vital capacity and peak expiratory flow) among patients with COPD.[35 36] Second, Shephard and Shek[37] reported that cold air could directly induce bronchoconstriction, leading to excessive dyspnoea in patients with COPD. Third, cold exposure may suppress the immune response and increase susceptibility to viral infections in humans.[38] Meanwhile, the transmission efficiency of the influenza virus is inversely correlated with ambient temperature.[39] Fourth, cold temperature may provoke airway inflammation and mucin hypersecretion in airway epithelium, which results in COPD morbidity and mortality by blocking airways and causing recurrent infections.[40 41]

We found that the CRR values of AECOPD hospitalisations increased with longer duration and higher intensity of cold spells. Some prior studies had similar findings,[20 21] indicating that both the duration and the intensity affect the health risks of cold spells. According to the minimum Q-AIC value criterion, the optimal definition of cold spells was daily average temperature at or below the 10th percentile for 4 or more consecutive days. Compared with other definitions, this one had lower intensity and longer duration, and the most significant single-day lagged effect of cold spells on AECOPD hospitalisations appeared earliest (lag0). Our results agree with a study in Porto showing that moderately low temperature for long periods contributed to COPD exacerbation greater than extremely low temperature with shorter-lasting days.[15] However, studies on some other diseases defined the optimal cold spell with the threshold at the fifth percentile or of at least 2 days' duration,[21 32] indicating different definitions may apply to different outcomes. In addition, some studies reported the temporal changes in people's adaption capacities to cold spells during recent decades under climate change.[5 42] Overall, more effective cold spell definitions and warning systems adapted to regional climate, specific diseases and dynamic changes of the population's sensitivity should be further studied and implemented in the future.

In the subgroup analyses based on the optimal cold spell definition, we found that the effects of cold spells on AECOPD hospitalisations were more significant in the

**Table 4** Cumulative relative risk of cold spells† on AECOPD hospitalisations stratified by gender and age in Beijing, 2012–2016

| Subgroup | CRR (95% CI) | | | |
| --- | --- | --- | --- | --- |
| | Lag0 | Lag0–7 | Lag0–14 | Lag0–21 |
| Male | 1.042 (1.011 to 1.074)* | 1.243 (1.123 to 1.375)* | 1.316 (1.173 to 1.477)* | 1.342 (1.136 to 1.586)* |
| Female | 1.041 (1.005 to 1.077)* | 1.257 (1.119 to 1.411)* | 1.383 (1.215 to 1.574)* | 1.476 (1.211 to 1.783)* |
| Age <65 | 1.017 (0.972 to 1.064) | 1.120 (0.963 to 1.303) | 1.159 (0.977 to 1.376) | 1.107 (0.862 to 1.422) |
| Age ≥65 | 1.046 (1.017 to 1.077)* | 1.275 (1.158 to 1.404)* | 1.382 (1.240 to 1.540)* | 1.456 (1.244 to 1.705)* |

*P<0.05.

†The optimal cold spell definition was daily mean temperature ≤10th percentile (–6℃) for at least 4 consecutive days during the study period.

AECOPD, acute exacerbation of chronic obstructive pulmonary disease; CRR, cumulative relative risk.

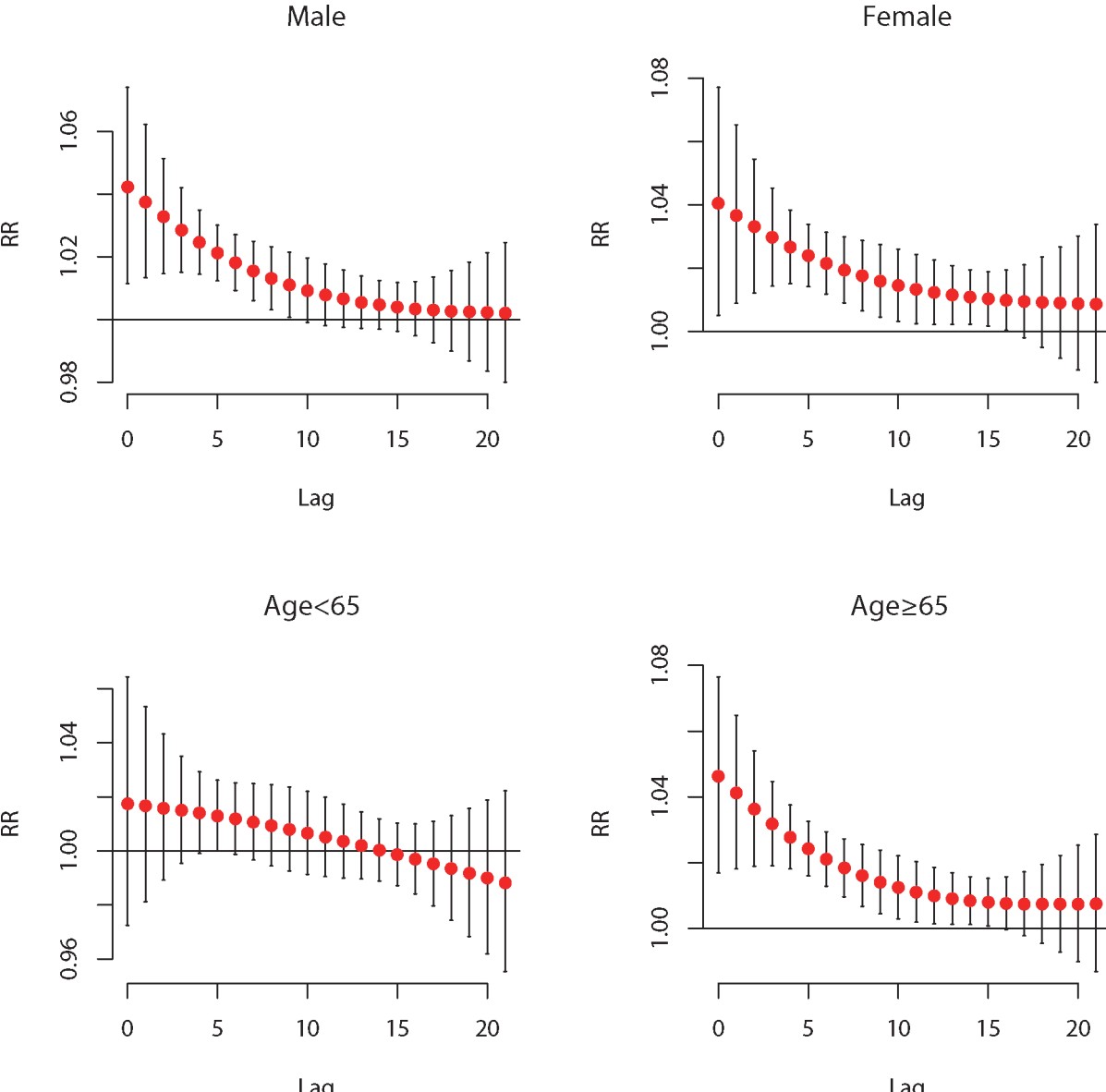

**Figure 2** Lag–response relationships between cold spells and AECOPD hospitalisations stratified by gender and age in Beijing, 2012–2016. AECOPD, acute exacerbation of chronic obstructive pulmonary disease; RR, relative risk.

elderly (aged ≥65 years) than people aged 0–64 years. This finding is consistent with previous studies.[4 12 43] The reasons may point to reduced thermoregulatory ability, higher prevalence of chronic diseases[20 34] and impaired immunity[44] in the elderly. Note that ageing is one of the risk factors of COPD, and most patients in our study were over 64, giving more power to achieve statistical significance. However, some studies showed the opposite results.[6] It was speculated that the younger tend to spend more time outdoors, increasing their opportunities for exposure to extremely cold temperature. In terms of gender, we found similar impacts of cold spells among male and female patients, which corresponds with previous studies.[4 12 43]

Our study has significant public health implications for local prevention of AECOPD, development of early warning systems and rational allocation of medical and health resources to mitigate the COPD burden caused by cold spells. We showed substantial effects of cold spells on the risk of AECOPD hospitalisations with a lagged effect of about 3 weeks, urging for effective and practical guidelines for preventions, particularly for patients with COPD during cold spells in China. Both the government and individuals should take practical actions. The meteorological departments should improve early warning systems with timely forecast and publication of extremely cold events. Moreover, the government should exert great efforts to raise public awareness of the health hazards of cold spells and ensure adequate public and medical services coping with cold spells.[4] As for individuals, it has been reported that staying indoor and wearing warm clothing could reduce mortality in extremely cold weather.[45] Tseng *et al*[12] suggested that patients with COPD who received inhaled medicine were less affected by cold

temperature-related COPD exacerbation. Therefore, keeping warm, minimising outdoor activities and taking medications regularly are vital measures to fight against cold spells for individuals with COPD, especially for the elderly.

The main strengths of our study are as follows: First, to the best of our knowledge, this is the first study to investigate the relationship between cold spells and AECOPD hospitalisations in China. Second, we controlled air quality and influenza epidemics as confounding factors, which were not included by some previous studies. Lee *et al*[13] have reported that a higher influenza virus detection rate was correlated with AECOPD.[14] A previous study from Beijing has found significant associations between short-term exposures to air pollution and hospitalisations for AECOPD.[23] Existing literature has also shown that both air pollution and influenza epidemics could contribute to cold-related health effects.[46 47] Moreover, air pollutants may interact with viral infections to precipitate AECOPD rather than acting alone. Feng *et al*[47] reported that ambient particulate matter with aerodynamic diameter <2.5 μm was associated with risk of influenza-like illness in Beijing during the influenza season. Further research on the interactions among cold spells, air pollution and influenza on AECOPD is therefore needed. Third, we identified the elderly to be more vulnerable to cold spells by stratified analyses, guiding more targeted prevention strategies.

The study also has several limitations. First, as an ecological study, the association between cold spells and AECOPD hospitalisations does not imply causality. Second, different socioeconomic status or other factors at an individual level might be confounding factors and were not considered in the association. Third, the meteorological variables and AQI were all from monitoring stations, not reflecting the individual level of exposures. Moreover, people are more likely to stay indoors with heating systems during extremely cold days in northern China, so inevitable exposure measurement errors may lead to bias. Fourth, due to the limited availability of local data, the positive rates of influenza isolations were from the northern part of China, but not only Beijing. Further studies with local influenza data included as a continuous variable in the model are warranted. Lastly, the data from only one city weaken the extrapolation validity of the study.

## CONCLUSION

Our study demonstrates that short-term exposure to cold spells is associated with an increased risk of AECOPD hospitalisations. The cumulative effects increased with the intensities and the durations of cold spells. The elderly are more vulnerable to AECOPD hospitalisations during cold spell periods. These findings provide scientific foundations for comprehensive public health strategies to reduce cold spell-related AECOPD hospitalisations in Beijing, China.

**Author affiliations**
¹Department of Medicine, Peking Union Medical College Hospital, Peking Union Medical College and Chinese Academy of Medical Sciences, Beijing, China
²Department of Physiotherapy, Peking Union Medical College Hospital, Peking Union Medical College and Chinese Academy of Medical Sciences, Beijing, China
³Department of Biochemistry, The State Key Laboratory of Medical Molecular Biology, Institute of Basic Medical Sciences, Chinese Academy of Medical Sciences, Peking Union Medical College, Beijing, China

**Contributors** ZF obtained the original data and funding. ZF, YL and YC designed the study. DK, XL, JF, YoZ and ZC preprocessed the data. YL, YC and YaZ analysed the data. YL and YC drafted the manuscript. XZ, KX, CJ and ZF reviewed and edited the manuscript. All authors have read and approved the final manuscript. ZF is the study guarantor.

**Funding** This work was supported by the National Key Research and Development Plan (2017YFC0211703), National 973 Project (2015CB553400), Chinese Academy of Medical Sciences (CAMS) Initiative for Innovative Medicine (CAMS 2016ZX310181-5/4, CAMS 2017-I2M-2-001), and National Natural Science Foundation (91643208, 41450006).

**Competing interests** None declared.

**Patient consent for publication** Not required.

**Ethics approval** This study was approved by the ethical review committee of Peking Union Medical College Hospital.

**Provenance and peer review** Not commissioned; externally peer reviewed.

**Data availability statement** All data relevant to the study are included in the article or uploaded as supplemental information. No additional data are available.

**ORCID iDs**
Jia Fu http://orcid.org/0000-0003-1432-0323
Zhongjie Fan http://orcid.org/0000-0002-2732-0659

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
