## [Reviewer comments · BMJ Open]

ARTICLE DETAILS

TITLE (PROVISIONAL)	The short-term effects of cold spells on hospitalizations for acute exacerbation of chronic obstructive pulmonary disease: a time-series study in Beijing, China
AUTHORS	Liu, Yanbo; Chen, Yuxiong; Kong, Dehui; Liu, Xiaole; Fu, Jia; Zhang, Yongqiao; Zhao, Yakun; Chang, Zhen'ge; Zhao, Xiaoyi; Xu, Kaifeng; Jiang, Chengyu; Fan, Zhongjie

VERSION 1 – REVIEW

REVIEWER	Prof. Dr. med. Christian Witt, Senior-Professor Dept. of Physiology, Charité - University Medicine Berlin, Germany Sauerbruchweg 3 10098 Berlin
REVIEW RETURNED	22-Jun-2020

GENERAL COMMENTS	Major comments: The investigation of COPD patients, exposed to cold air is already performed in the past until to clinical test, established as Cold Air Bronchial Provocation test. The relationship to the pathophysiology of the disease is accepted in general, in contrast to the heat exposure (e.g. Gasparini et al. Lancet). This manuscript reports both, the local situation for China/Beijing and global the effect of cold spells in vulnerables patient groups, The relation to the climate change is less than the heat exposure of COPD patients. Express in Detail please, what is really new in this work, what is beyond the present knowledge status? It is of interest for clinicians or physicians, but to improve the manuscript into that direction, it needs more translational effort. The statistical methods should be reviewed in addition to by a specialist for current statistic analyses. My last point is focussed on the title. Why are you insist on short-term effects ? effects only ? Minor comments: The approach was started with the main questions about cold (temperature), Air Quality and Influenza status, but insufficient explanations were done about air quality and influenza and their relationship to each other. We know, that this is important to investigate cold periods and vulnerability.
---

REVIEWER	Chin Kook Rhee Seoul St. Mary's Hospital, College of Medicine, The Catholic University of Korea, Republic of Korea
REVIEW RETURNED	02-Jul-2020

GENERAL COMMENTS	In this study, the authors aimed to explore the relationship between cold spells and COPD hospitalization in Beijing. Cold spells were
--

	associated with increased risk of hospitalization. The cumulative effects at lag0-21 increased with the intensities and the durations of cold spells. This is an interesting study and the manuscript is well written. The result is clear and informative. Major comments  1. The limitation of this study is lack of novelty. The authors stated that few studies have been carried out on the very issue. However, there have been some previous studies that showed similar results (PLoS One 2013; 8: e57066., Thorax 2018; 73: 951-958., Sci Rep 2019; 9: 6679.). 2. The working definition of COPD hospitalization should be validated. The working definition of COPD hospitalization in this study is J44 as primary discharge diagnosis. Is this definition validated? Please provide the reference. COPD patients can be hospitalized due to causes other than respiratory events. For example, when COPD patients had fracture, they may admit the hospital. These kinds of events have nothing to do with cold spell. Can the authors select hospitalization due to respiratory causes? 3. The COPD hospitalization data is collected from Beijing Municipal Health Commission Information Center. Is this data reliable? It will be better if the authors refer previous studies using this database. 4. Page 6, 129th line. Influenza is one of the important factors associated with COPD hospitalization. What was the source of the influenza epidemics? Was it national level (entire China) or local level (Beijing) data? How influenza data was generated? 5. The authors adjusted influenza infection as binary variable. However, it should be adjusted as continuous (quantitative) variable (Sci Rep 2019; 9: 6679.). Minor comment  1. Table 1 is hard to understand. What do “Mean (SD) and Min~Max” mean for COPD cases? Do they mean the number of hospitalizations/day? Same in gender and age.
--	---

VERSION 1 – AUTHOR RESPONSE

Reviewer#1:

- Major Comment 1: The investigation of COPD patients, exposed to cold air is already performed in the past until to clinical test, established as Cold Air Bronchial Provocation test. The relationship to the pathophysiology of the disease is accepted in general, in contrast to the heat exposure (e.g. Gasparini et al. Lancet). This manuscript reports both, the local situation for China/Beijing and global the effect of cold spells in vulnerables patient groups, The relation to the climate change is less than the heat exposure of COPD patients. Express in Detail please, what is really new in this work, what is beyond the present knowledge status? It is of interest for clinicians or physicians, but to improve the manuscript into that direction, it needs more translational effort. The statistical methods should be reviewed in addition to by a specialist for current statistic analyses.

Response:

Thank you for your informative questions. For the novelty of our work, we think there are several points worth mentioning.

Firstly, we agree that the relationship between cold temperatures and COPD has been extensively researched and translated into clinical practice. However, our work focused on cold spells, which are defined as periods of consecutive days with extremely low temperatures. The health effects of prolonged periods of extreme temperatures may increase due to continuous days of exposures compared to linear changes of daily temperature¹. Yet, few studies have focused on the effects of cold spells on AECOPD hospitalizations. Secondly, as the most populated country in the world, China has a relatively high burden of COPD, while no specific studies on cold spells and COPD have been conducted in China. Heterogeneity in climate, healthcare system and socio-economic status across regions may limit the extrapolation of the research results, urging for more studies conducted in different climate regions in this field. Thirdly, we not only reported on the relationship between the two but also explored the optimal definition of cold spells in the region, which is helpful for the development of local early warning systems and rational allocation of medical and health resources for cold spells.

To further illustrate the novelty and significance of our work, we have revised the related sections in the introduction and discussion part (the fifth paragraph, Page 11, Line 316-318). Please see the revised manuscript for details.

- Major Comment 2: My last point is focussed on the title. Why are you insist on short-term effects? effects only?

Response:

The reason why we focused on the short-term effects is based on the following two points. Firstly, in environmental epidemiology, time-series models are used to characterize the variation in environmental conditions (e.g., air pollution, ambient temperature and other meteorological factors), also to investigate the relationship of short-term exposures and health outcomes². Previous studies concerning the health effects of cold spells or other environmental factors with time-series methods also emphasized short-term effects in their titles³⁻⁵. Secondly, your question did prompt us to think about how we can communicate our work more precisely. Another reviewer has pointed out that our definition of COPD hospitalizations (ICD-10 code: J44) should be validated. We looked into the original data and believed that a more precise description of our cases should be hospitalizations for acute exacerbations of COPD (AECOPD), whereas stable COPD patients tend to get treatment in the out-patient settings. It is of greater significance to explore the acute effects of cold spells on AECOPD to guide early warning systems and policymaking. We have changed the description of the outcome from "COPD hospitalizations" to "AECOPD hospitalizations" throughout the manuscript including the title.

We can't quite understand the meaning of "effects only?" at the end of the comments. We would appreciate if you can provide more details.

- Minor Comment: The approach was started with the main questions about cold (temperature), Air Quality and Influenza status, but insufficient explanations were done about air quality and influenza and their relationship to each other. We know, that this is important to investigate cold periods and vulnerability.

Response:

These are very insightful comments! Both air quality and influenza have been considered as confounding factors in our model. Since we mainly focus on the cold spells, we did not go in depths to explore the effects of air pollution and influenza as well as the potential interaction between them. We added references supporting the associations of air quality and influenza with AECOPD and emphasized the need for future studies on the possible interactions of cold spells, air pollution and

influenza in the discussion part. Please see Page 11-12, Line 336-344 in the revised manuscript for details.

Reviewer#2:

- Major Comment 1: In this study, the authors aimed to explore the relationship between cold spells and COPD hospitalization in Beijing. Cold spells were associated with increased risk of hospitalization. The cumulative effects at lag0-21 increased with the intensities and the durations of cold spells. This is an interesting study and the manuscript is well written. The result is clear and informative.

Response:

Thank you for the approval of our work.

- Major Comment 2:

The limitation of this study is lack of novelty. The authors stated that few studies have been carried out on the very issue. However, there have been some previous studies that showed similar results (PLoS One 2013; 8: e57066., Thorax 2018; 73: 951-958., Sci Rep 2019; 9: 6679.).

Response:

We do appreciate these insightful comments. However, we believe the studies mentioned are somewhat different from our work and hereby further novelties of our work.

The second article (Thorax 2018; 73: 951-958.) was focused on the long-term effects of temperature variability in different seasons in Hong Kong on hospitalizations for respiratory diseases and suggested that wintertime temperature variability would increase the risk of incident respiratory diseases⁶. There are several differences between the paper and our work. Firstly, cold spells are defined as periods of continuously extremely low temperatures, which is different from temperature variability, defined as the standard deviation of the daily temperature. Secondly, the study population in our study was people of all ages while the paper focused on the elderly (≥ 65 years). Thirdly, Beijing was our study area with a subtropical monsoon climate while the study area in the paper was Hong Kong with a subtropical monsoon climate.

The other two papers (PLoS One 2013; 8: e57066., Sci Rep 2019; 9: 6679.) both focused on the effect of low temperatures on AECOPD, one conducted in Taiwan⁷ the other in South Korea⁸. Both indicated that lower daily temperatures were associated with higher risks of AECOPD. We agree that the relationship between cold temperatures and COPD has been extensively researched and added these them as references (See Reference[13], [14] in the revised manuscript). However, our study focused on cold spells, characterized by consecutive days of low temperatures. A previous study showed that heatwaves were associated with excesses of mortality above those expected from linear increments with increasing daily temperature¹. Therefore, we speculated that cold spells may be more detrimental than decreasing daily low temperatures. To the best of our knowledge, this is the first study conducted in China to explore the relationship between cold spells and AECOPD hospitalizations. Also, we explored the optimal definition of cold spells in terms of intensity and duration, which is helpful for the development of local warning systems and efficient allocation of healthcare resources for cold spells.

To further explain the novelty and significance of our work, we have revised the related sections in the introduction and discussion part (the fifth paragraph, Page 11, Line 316-318). Please see the revised manuscript for details.

- Major Comment 3:

The working definition of COPD hospitalization should be validated. The working definition of COPD hospitalization in this study is J44 as primary discharge diagnosis. Is this definition validated? Please provide the reference. COPD patients can be hospitalized due to causes other than respiratory

events. For example, when COPD patients had fracture, they may admit the hospital. These kinds of events have nothing to do with cold spell. Can the authors select hospitalization due to respiratory causes?

Response:

It is a great point! We had vigorous discussions within our team on this issue before we sent out the manuscript. When it comes to screening patients of COPD, there is no single best way in the field. Some researchers used J41-J44 as criteria. A previous study carried out on the air quality and hospitalizations for acute exacerbations of COPD (AECOPD) in Beijing also applied the selection criterion of code J44 as primary diagnosis⁹. In the ICD-10 coding system, chronic bronchitis and emphysema without signs of obstruction are coded J41-J43, which we believe don't meet COPD diagnosis and can't be included in the study. Moreover, it is commonly believed that in the healthcare system of Beijing and China, COPD patients only get admitted in the event of acute exacerbations, so we decided to include all patients in the category of J44, which includes COPD with pulmonary infection, acute COPD exacerbations, etc. As for the fracture scenario, all cases in our study were screened by the first discharge diagnosis. If a patient with COPD was admitted due to a fracture, his/her first discharge diagnosis would be fracture rather than COPD, thus will not be included in our study.

For the comment, we have changed the description of the outcome from "COPD hospitalizations" to "AECOPD hospitalizations" throughout the manuscript. Besides, the previous study using the same definition of AECOPD hospitalizations mentioned above was cited in the methods part (Please see Page 5, Line 121-123 and Reference[23] in the revised manuscript for details).

- Major Comment 3:

The COPD hospitalization data is collected from Beijing Municipal Health Commission Information Center. Is this data reliable? It will be better if the authors refer previous studies using this database.

Response:

The data source is the most reliable we can find and there have been many studies published using the same database. The data center was formerly known as the Beijing Public Health Information Center and was officially renamed the Beijing Municipal Health Commission Information Center in July 2019 (<http://www.phic.org.cn/>).

In the methods part, we cited two previous studies using this database and added the website and a brief description. Please see Page 4-5, Line 114-118 and Reference[22], [23] in the revised manuscript for details.

- Major Comment 4:

Page 6, 129th line. Influenza is one of the important factors associated with COPD hospitalization. What was the source of the influenza epidemics? Was it national level (entire China) or local level (Beijing) data? How influenza data was generated?

Response:

We feel sorry that we did not provide enough information about influenza data. Our influenza data were collected from the Chinese National Influenza Center (CNIC) (<http://www.chinaivdc.cn/cnic/>). The center monitors the activity of seasonal influenza viruses in China and reports weekly positive rates of influenza isolations and influenza-like cases (ILI) percentage (%) in the northern and southern parts separately. Beijing is located in the north of China. Since we were unable to get the influenza data specifically for Beijing, we used the influenza data from the northern part of China. To further evaluate whether the data applied to Beijing, we referred to an article in Chinese from the Beijing Center for Disease Prevention and Control. The article displayed a graph of the weekly distribution of ILI% and the positive detection rate of influenza virus during 2012-2015 but no raw data¹⁰. We found that the trend of positive detection rates of influenza virus from the northern part of China was roughly

consistent with that reported in the literature during the same period. Hence, we think it's reasonable to use the influenza data of northern regions to reflect the tendency of influenza epidemics in Beijing during the study period. Finally, we defined the onset of influenza epidemics (a binary variable representing days with relatively high influenza episodes) as when the proportion of isolates positive for influenza in any given week exceeded 30% of the maximum weekly positive isolation rate in the whole surveillance season (Influenza surveillance season was defined from the 27th week of the previous year to the 26th week of the following year) in northern China.

To further clarify the source of the influenza epidemics, we revised the relevant section in the methods part. Please see Page 5, Line 133-143 in the revised manuscript for details.

- Major Comment 5:

The authors adjusted influenza infection as binary variable. However, it should be adjusted as continuous (quantitative) variable (Sci Rep 2019; 9: 6679.).

Response:

The article provided by the reviewer is of great help to us, which stated that the influenza is one of the important risk factors associated with AECOPD hospitalizations. We agreed that a better model fit can potentially be achieved if we adjusted influenza as a continuous variable. However, due to the limited availability of local data, the influenza data was from the northern part of China. The data could reflect the tendency of influenza epidemics in Beijing but not the accurate local level. The difference in data measurement period (e.g., daily AECOPD hospitalizations and weekly influenza data) should also be considered. Since we did not focus on the effects of influenza but just consider it as one of the confounding factors, we believe that treating influenza epidemics as a binary variable to mark the influenza peak period is reasonable. Some previous studies with time-series methods also adjusted influenza status as a binary variable^{11,12}, and we added two as references in the methods part (See Reference[25], [26] in the revised manuscript for details). We have tried to adjust influenza data as a continuous variable in our model. The overall trend of the effects of cold spells remained the same with only small variations of the values of relative risks.

We agree with the reviewer's suggestions and therefore consider our influenza data should be mentioned as a limitation. We added the relevant section in the description of the study limitations in the discussion part. Please see Page 12, Line 354-356 in the revised manuscript for details.

- Minor Comment:

Table 1 is hard to understand. What do "Mean (SD) and Min~Max" mean for COPD cases? Do they mean the number of hospitalizations/day? Same in gender and age.

Response:

Thank you for your question. We apology for any confusion. Table 1 shows the statistic summary of daily numbers of AECOPD hospitalizations (total population and the subpopulations stratified by gender and age) as well as the daily data of meteorological factors and air quality during the study period. Mean (SD) represents the average number and standard deviation (SD) of daily counts of AECOPD hospitalizations during the study period. Min and Max represent the minimum and the maximum number of daily AECOPD hospitalizations during the study period, respectively. We have revised Table 1 to avoid misunderstandings. Please see Page 17 in the revised manuscript for details.

We also checked our data again and found some mistakes in Table 3, which have been corrected in the revised manuscript. Please see Page 19 in the revised manuscript for details.

VERSION 2 – REVIEW

REVIEWER	christian witt Charité - University Hospital Berlin, Germany
REVIEW RETURNED	19-Oct-2020

GENERAL COMMENTS	s. the review
---------------

REVIEWER	Chin Kook Rhee Seoul St. Mary's Hospital, The Catholic University of Korea
REVIEW RETURNED	27-Sep-2020

GENERAL COMMENTS	The authors have adequately revised manuscript. I have no further comment.
--